# BEHIND RoPE: HOW DOES CAUSAL MASK ENCODE POSITIONAL INFORMATION?

## ABSTRACT

While explicit positional encodings such as RoPE are a primary source of positional information in Transformer decoders, the causal mask also provides positional information. In this work, we prove that the causal mask can induce position-dependent patterns in attention scores, even without parameters or causal dependency in the input. Our theoretical analysis indicates that the induced attention pattern tends to favor nearby query-key pairs, mirroring the behavior of common positional encodings. Empirical analysis confirms that trained models exhibit the same behavior, with learned parameters further amplifying these patterns. Notably, we found that the interaction of causal mask and RoPE distorts RoPE's relative attention score patterns into non-relative ones. We consistently observed this effect in modern large language models, suggesting the importance of considering the causal mask as a source of positional information alongside explicit positional encodings.

## 1 INTRODUCTION

Transformer decoders (Vaswani et al., 2017) with rotary positional embeddings (RoPE) (Su et al., 2024) have been widely adopted in recent large language models (LLMs) (Grattafiori et al., 2024; Abdin et al., 2024; Yang et al., 2025a). The way positional information is provided to a model is known to be closely tied to model performance (Dufter et al., 2022) and its length generalization ability (Zhao et al., 2024). Consequently, recent work has sought to improve LLMs, in terms of both LLM performance (Barbero et al., 2025) and length generalization (Peng et al., 2024; Chen et al., 2023; Liu et al., 2024), by analyzing and modifying RoPE.

However, these models contain another source of positional information: the causal mask. It is commonly viewed as a mechanism that blocks access to future tokens, but it also provides positional information. Although the exact mechanism remains unclear (Zuo et al., 2025), recent studies have shown that models without explicit positional encodings can still model sequential data and even achieve performance comparable to models with RoPE (Haviv et al., 2022; Kazemnejad et al., 2023). Similar to RoPE, **analyzing how the causal mask encodes positional information and its properties** is crucial for understanding model behavior, as well as its implications for performance and length generalization.

Although several recent studies have attempted to analyze how the causal mask encodes positional information (Haviv et al., 2022; Chi et al., 2023; Kazemnejad et al., 2023), its exact nature remains unclear (Zuo et al., 2025). Thus in this paper, we first prove that even without parameters, causal input dependencies, or a feedforward network, the causal mask can induce position-dependent patterns in attention scores (Figure 1). These patterns consistently favor closer keys to each query, assigning them higher attention scores. This behavior closely resembles that of many explicit positional encoding schemes (Press et al., 2022; Su et al., 2024; Vaswani et al., 2017).

Through empirical analysis, we then demonstrate that our explanation aligns well with practical outcomes and uncovers several useful characteristics. First, by simulating a Transformer decoder without parameters and without explicit positional encoding, we confirm that our explanation accurately captures how positional information emerges. We also find that the resulting position-dependent attention patterns exhibit properties similar to positional encodings, yet quite differ qualitatively from both conventional absolute and relative forms. Furthermore, training a Transformer decoder

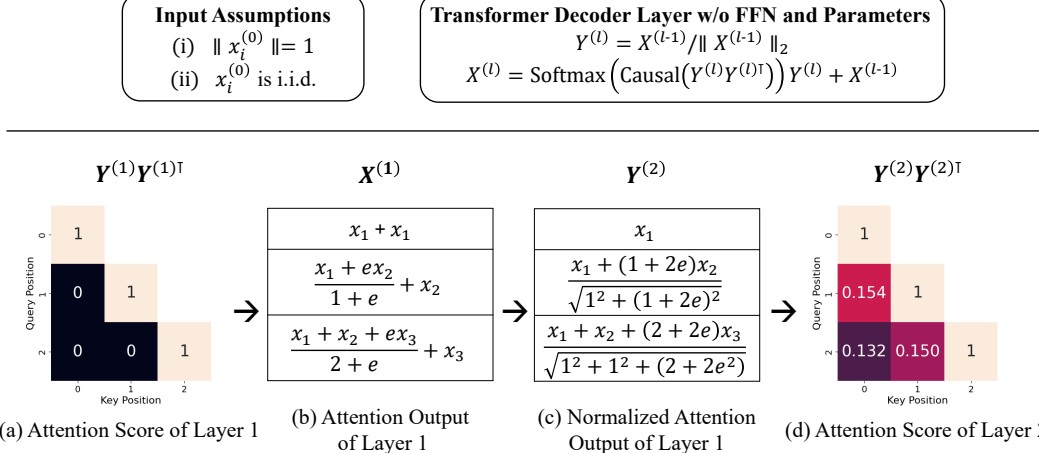

(a) Attention Score of Layer 1

(b) Attention Output of Layer 1

(c) Normalized Attention Output of Layer 1

(d) Attention Score of Layer 2

Figure 1: Causal mask induces positional information even in the absence of causal input dependencies, feed-forward networks, or parameters. (a) With the input assumption, the first-layer attention scores collapse to 1 on the diagonal and 0 elsewhere. (b) The attention output is then computed as a weighted sum of the $x_i^{(0)}$ (we omit the superscript $^{(0)}$ and simply write $x_i$ in place of $x_i^{(0)}$). (c) After $\ell_2$ normalization, (d) subsequent-layer attention scores reveal a clear position-dependent pattern, assigning higher weights to nearby query–key pairs, a behavior similar to common positional encodings.

on a web corpus without explicit positional encodings shows that the emergence of positional information in practice is consistent with our explanation. However, while the underlying mechanism matches, we observe that in the real model the position-dependent attention patterns are strongly influenced by the learned parameters.

In addition to our theoretical and empirical analysis on the properties of the causal mask, we study how it interacts with RoPE inside modern LLMs, which typically use both together. Simulations of a parameter-free Transformer show that when combined with RoPE, the causal mask distorts RoPE's relative attention pattern into a non-relative one. Our analysis further reveals that this non-relative pattern arises only in the presence of the causal mask. We consistently observe this phenomenon at a non-negligible scale in modern LLMs that use RoPE, including Llama-3.1-8B (Grattafiori et al., 2024), Phi-4 (Abdin et al., 2024), and Qwen3-8B (Yang et al., 2025a).

Our contributions are as follows:

- We prove that the causal mask can induce position-dependent patterns in attention scores, even in the absence of parameters, causal input dependencies, or a feedforward network.
- Through empirical analysis, we demonstrate that our explanation accounts for the behavior of Transformer decoders without explicit positional encoding.
- We show that the causal mask biases RoPE's relative attention pattern toward a non-relative one, and we observe this bias in modern LLMs.

These results suggest that future research on positional information in Transformer decoders should account for the joint effects of both RoPE and the causal mask.

## 2 RELATED WORKS

A typical way to inject positional information into Transformers is through positional encodings. They generally encourage higher attention scores for query–key pairs that are closer together (Press et al., 2022; Su et al., 2024; Vaswani et al., 2017). Broadly, there are two types of positional encodings. Absolute positional encodings assign information based on each token's fixed position in the sequence, such as sinusoidal encoding and learnable absolute positional encodings (Vaswani et al., 2017). Relative positional encodings represent positions based on the distance between tokens, such

as T5 relative PE (Raffel et al., 2020), ALiBi (Press et al., 2022), and RoPE (Su et al., 2024). Among these, RoPE has become widely adopted in recent LLMs (Abdin et al., 2024; Grattafiori et al., 2024; Yang et al., 2025a), and its properties have inspired methods that improve both language modeling performance (Barbero et al., 2025; Yang et al., 2025b) and length generalization (Peng et al., 2024; Chen et al., 2023; Liu et al., 2024).

Although positional encoding is typically considered the sole source of positional information in Transformer decoders, recent work has shown that the causal mask can also play this role. Haviv et al. (2022) first demonstrated that a Transformer decoder can model natural language without explicit positional encodings, achieving performance comparable to the model with RoPE. Because the causal mask is typically viewed simply as a mechanism for blocking access to future tokens, several works have attempted to uncover how it encodes positional information. In the same work, Haviv et al. (2022) hypothesized that the causal mask enables counting of predecessor tokens, though they did not provide a proof. They also suggested that the positional information induced by the causal mask resembles absolute positional encoding. Kazemnejad et al. (2023) later proved that the causal mask can represent both absolute and relative encodings under a specific parameter configuration. Also, they showed that the attention pattern without positional encoding is closely resemble to those of T5's relative positional embeddings. Chi et al. (2023) offered the first mathematical explanation, showing that the causal mask increases the variance of hidden states with token position. However, they did not clarify how Transformers exploit variance from a single hidden state. More recently, Zuo et al. (2025) empirically showed that nearby hidden states exhibit higher cosine similarity than distant ones, and that this tendency is much stronger than variance change. Extending this line of work, we explain how the causal mask can encode positional information, thereby justifying the pattern observed by Zuo et al. (2025). We also show that the positional information from causal mask behavior quite differs from both absolute and relative positional encodings.

## 3 THEORETICAL ANALYSIS

### 3.1 PRELIMINARIES

Let the input token embeddings be $X^{(0)} = [x_1^{(0)}, \cdots, x_n^{(0)}] \in \mathbb{R}^{n \times d}$, where $n$ is the number of input tokens and $d$ is the model hidden size. Superscripts indicates layers; when clear from context, we omit them. Formally, a single-head, pre-LN (Xiong et al., 2020) Transformer decoder layer without bias is a function $f : \mathbb{R}^{n \times d} \to \mathbb{R}^{n \times d}$ with $X^{(l)} = f^{(l)}(X^{(l-1)})$ is defined as:

$$Y^{(l)} = \text{LayerNorm}(X^{(l-1)}), \quad Q = YW_Q, \quad K = YW_K, \quad V = YW_V,$$

$$A = \text{Softmax}(\text{Causal}(\frac{QK^\top}{\sqrt{d}})), \quad O = (AV)W_O + X^{(l-1)},$$

$$X^{(l)} = \text{FFN}(\text{LayerNorm}(O^{(l)})) + O^{(l)}$$

Where $W_Q, W_K, W_V, W_O \in \mathbb{R}^{d \times d}$ are parameters, and operation $\text{Causal}(\cdot)$ applies a strictly upper-triangular mask to prevent attention to future positions.

### 3.2 HOW DOES CAUSAL MASK ENCODE POSITIONAL INFORMATION?

Here, we show that the causal mask can induce a position-dependent pattern in attention score, even when the input sequence has no causal dependency, no parameters, and no feed-forward module. In addition, we show the pattern allocates higher attention scores to closer query-key pairs, akin to the behavior of typical positional encodings (Press et al., 2022; Su et al., 2024; Vaswani et al., 2017). Figure 1 sketches the high-level mechanism by which the causal mask encodes positional information.

To simplify the derivation, we employ $\ell_2$ normalization (without the $\sqrt{d}$ term) as the normalization technique, and later show that LayerNorm (with the $\sqrt{d}$ term) exhibits analogous behavior. We assume each input embedding vector $x_i^{(0)}$ has unit norm, and for $i \neq j$, $\mathbb{E}(\langle x_i^{(0)}, x_j^{(0)} \rangle) = \alpha$. Note that this assumption does not impose causal structure on the inputs and includes an i.i.d. case ($\alpha = 0$). While $\alpha = 0$ is sufficient for our core claim, we allow $0 \leq \alpha < 1$ to better explain the

trained model's behavior (Section 4.2). Under these simplifications, a single layer $f(X)$ acts as:

$$f(X) = \text{Softmax}(\text{Causal}(YY^\top))Y + X, \quad Y = \text{L2Norm}(X), \tag{1}$$

where L2Norm denotes row-wise $\ell_2$ Normalization. The operator Causal$(\cdot)$ applies the strictly upper triangular mask so that a query at position $i$ only attends to keys at position $j \leq i$, and Softmax is taken row-wise. Formally, our goal is to show that the pairwise inner product after normalization $\langle y_i^{(2)}, y_j^{(2)} \rangle$ is a function of the indices $i$ and $j$ (i.e. not constant across positions).

Since each input has unit norm, $\langle x_i^{(0)}, x_i^{(0)} \rangle = 1$, we have $Y^{(1)} = X^{(0)}$. The Gram matrix of inner products and those after applying the causal mask is:

$$(Y^{(1)}Y^{(1)\top})_{i,j} = \begin{cases} 1 & (i = j) \\ \alpha & (i \neq j) \end{cases} \tag{2}$$

$$\text{Causal}(Y^{(1)}Y^{(1)\top})_{ij} = \begin{cases} 1 & (i = j) \\ -inf & (i < j) \\ \alpha & (i > j) \end{cases}$$

The row-wise softmax then gives

$$\text{Softmax}(\text{Causal}(Y^{(1)}Y^{(1)\top}))_{ij} = \frac{f(i,j)}{e + (i-1)e^\alpha} \quad \text{where } f(i,j) = \begin{cases} e & (i = j) \\ 0 & (i < j) \\ e^\alpha & (i > j). \end{cases}$$

Accordingly,

$$x_i^{(1)} = \frac{\sum_{k=1}^{i} f(i,k)x_k^{(0)}}{e + (i-1)e^\alpha} + x_i^{(0)} = \frac{(2e + (i-1)e^\alpha)x_i^{(0)} + \sum_{k=1}^{i-1} e^\alpha x_k^{(0)}}{e + (i-1)e^\alpha}$$

We first compute the raw inner product $\langle x_i^{(1)}, x_j^{(1)} \rangle$, and then normalize by $||x_i^{(1)}||_2$ and $||x_j^{(1)}||_2$ to obtain $\langle y_i^{(2)}, y_j^{(2)} \rangle$.

$$\langle x_i^{(1)}, x_j^{(1)} \rangle = \frac{\big((2e + (i-1)e^\alpha)x_i^{(0)} + \sum_{k=1}^{i-1} e^\alpha x_k^{(0)}\big)\big((2e + (i-1)e^\alpha)x_j^{(0)} + \sum_{l=1}^{j-1} e^\alpha x_l^{(0)}\big)}{(e + (i-1)e^\alpha)(e + (j-1)e^\alpha)}$$

For $i > j$,

$$\langle x_i^{(1)}, x_j^{(1)} \rangle = \frac{\begin{aligned} &(2e + (i-1)e^\alpha)(2e + (j-1)e^\alpha)\langle x_i^{(0)}, x_j^{(0)} \rangle \\ &+ (2e + (i-1)e^\alpha)\sum_{\ell=1}^{j-1} e^\alpha \langle x_i^{(0)}, x_\ell^{(0)} \rangle \\ &+ (2e + (j-1)e^\alpha)\sum_{k=1}^{i-1} e^\alpha \langle x_k^{(0)}, x_j^{(0)} \rangle + \sum_{k=1}^{i-1}\sum_{\ell=1}^{j-1} e^{2\alpha}\langle x_k^{(0)}, x_\ell^{(0)} \rangle \end{aligned}}{(e + (i-1)e^\alpha)(e + (j-1)e^\alpha)}$$

$$= \frac{\begin{aligned} &(2e + (i-1)e^\alpha)(2e + (j-1)e^\alpha)\alpha + (2e + (i-1)e^\alpha)\,e^\alpha(j-1)\alpha \\ &+ (2e + (j-1)e^\alpha)\,e^\alpha\big(1 + (i-2)\alpha\big) + e^{2\alpha}(j-1)\big(1 + (i-2)\alpha\big) \end{aligned}}{(e + (i-1)e^\alpha)(e + (j-1)e^\alpha)}$$

$$= \frac{2\big(e + (j-1)e^\alpha\big)\Big(\alpha(2e + (i-1)e^\alpha) + e^\alpha\big(1 + (i-2)\alpha\big)\Big)}{(e + (i-1)e^\alpha)(e + (j-1)e^\alpha)}$$

$$= \frac{2\Big(2\alpha e + e^\alpha\big(1 + \alpha(2i-3)\big)\Big)}{e + (i-1)e^\alpha} = g(i)$$

For $||x_i^{(1)}||_2$,

$$||x_i^{(1)}||_2^2 = \frac{(2e + (i-1)e^\alpha)^2 + 2(2e + (i-1)e^\alpha)e^\alpha \alpha(i-1) + e^{2\alpha}(i-1)(1 + (i-2)\alpha)}{(e + (i-1)e^\alpha)^2}$$

$$= h(i)^2 \tag{3}$$

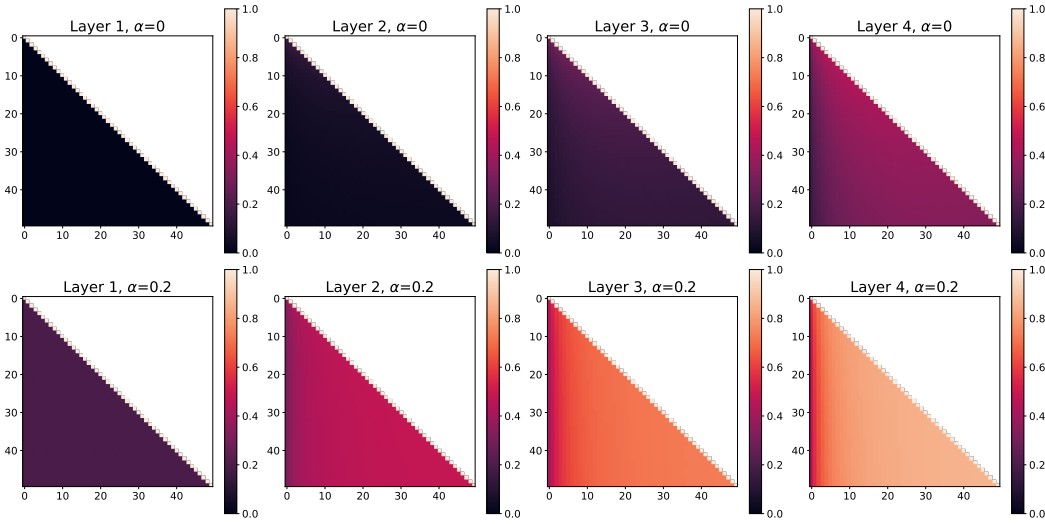

Figure 2: Simulation of a Transformer without parameter and explicit positional encoding results. We visualized averaged attention scores for each layer with $\alpha = 0$ and 0.2. The y-axis represents query indices, and the x-axis represents key indices.

Put everything together, we can get

$$\langle y_i^{(2)}, y_j^{(2)} \rangle = \frac{\langle x_i^{(1)}, x_j^{(1)} \rangle}{||x_i^{(1)}||_2 ||x_j^{(1)}||_2} = \frac{g(i)}{h(i)h(j)} \tag{4}$$

Thus, $\langle y_i^{(2)}, y_j^{(2)} \rangle$ is not constant across query-key indices $i, j$. This shows that the causal mask alone induces a position-dependent inner product structure in the normalized representations, even without any causal assumptions on input, without parameters, and without feed-forward network. In other words, causal mask itself can serve as a mechanism for encoding positional information.

Next, we show that the position-dependent attention pattern induced by the causal mask at the second layer behaves similar to typical positional encodings. Typical positional encodings infuse a bias into the attention score, making queries more strongly associated with nearby keys than with distant ones (Press et al., 2022; Su et al., 2024; Vaswani et al., 2017). Formally, we show the attention score in the second layer strictly increases on the key index $j \leq i$ over fixed query index $i$. From Equation 4, for such an $i$, the $g(i)/h(i)$ term becomes constant, so the score depends only on $1/h(j)$. We can compute $h(j+1) - h(j)$ and verify that $h(j)$ decreases strictly with $j$ (see Appendix A.1). As a result, the attention score at the second layer increases strictly with $j$, as long as $j < i$. In the case $j = i - 1$, since the inner product of two normalized vectors with different directions is always less than 1, we obtain

$$\langle y_i^{(2)}, y_{i-1}^{(2)} \rangle < \langle y_i^{(2)}, y_i^{(2)} \rangle = 1.$$

Therefore, for any fixed $i$, the attention score increases strictly with $j$ on the range $j \leq i$ in layer 2. In other words, closer keys receive higher scores, matching the behavior of common positional encodings. We empirically show the behavior of later layers in the following section.

## 4 EMPIRICAL ANALYSIS

### 4.1 TRANSFORMER SIMULATION WITHOUT PARAMETERS

We further examine the behavior of positional information from the causal mask through simulation of a Transformer without parameters and without positional encodings, as defined in Equation 1. Specifically, we sampled 50 vectors with $d = 64$ that satisfy Equation 2 in expectation. Each vector is generated by combining a shared Gaussian component with an independent Gaussian noise

component, ensuring the desired inner-product structure without introducing causal dependency. Figure 2 presents the simulated attention scores across layers for $\alpha = 0$ and $\alpha = 0.2$, averaged over 100,000 simulations. We also conducted the same experiment with $\ell_2$ Normalization replaced by LayerNorm, and observed similar tendencies, as confirmed in Appendix B.

First, consider the case $\alpha = 0$, corresponding to the first row of Figure 2. Under our standing assumption, the attention matrix of the first layer has ones on the diagonal and zeros elsewhere. In the second layer, a position-dependent pattern begins to emerge, consistent with our earlier derivations. Notably, the attention score strictly increases for $j \leq i$ with fixed $i$, consistent with our theoretical analysis. However, as shown in Figure 1, the exact across-position differences are still small at this stage, yielding only a faint pattern in the upper-left of the matrix. By layers three and four, the pattern becomes more pronounced, and also strictly increasing for $j \leq i$ with fixed $i$, resembling the behavior of common positional encodings. This attention pattern aligns with Figure 1 in Zuo et al. (2025), which illustrates that cosine similarity between hidden states in a Transformer decoder without positional encoding is higher for closer $i, j$ pairs. Our analysis explains this phenomenon, since cosine similarity is equivalent to the inner product after $\ell_2$ normalization.

Next, we examine the case $\alpha = 0.2$, illustrated in the second row of Figure 2. While a strict increase in the attention score for $j \leq i$ (with $i$ fixed) also occurred, the overall pattern is a bit different. At the second layer, the position-dependent pattern is much clearer compared to the $\alpha = 0$ case, but the pattern seems nearly independent of $i$. This effect occurs because, when $\alpha \neq 0$, the numerator and denominator share identical highest-order terms in $g(i)$, causing rapid convergence to a fixed value. The same saturation is more evident in the later layers.

In both cases, we observed that the attention scores within each off-diagonal band (i.e., excluding the main diagonal) were highly non-uniform. Since relative positional encodings inject information based on the token distance, attention scores within the diagonal are expected to remain uniform when the input is zero-mean Gaussian noise without learnable parameters (see Appendix Figure 10 for an example with RoPE). This discrepancy indicates that the causal mask behaves quite differently from relative positional encodings. On the other hand, Wang & Chen (2020) showed that absolute positional encodings, including sinusoidal and learnable embeddings, yield attention score heatmaps symmetric along the bottom-left to top-right axis. In the case of the causal mask, however, the heatmap does not satisfy such symmetry. Taken together, the behavior of positional information from the causal mask quite differs from both typical absolute and relative positional encodings.

## 4.2 ANALYSIS OF A TRAINED MODEL WITHOUT POSITIONAL ENCODING

We conducted an empirical study to examine whether similar attention patterns arise when training a Transformer decoder without positional encoding. To this end, we trained a model based on the Llama-3 architecture (Grattafiori et al., 2024) having 1.5B parameters (22 layers, hidden dimension 2048, head dimension 64) on 20 billion tokens from the Fineweb-Edu corpus (Penedo et al., 2024). The model was trained with an AdamW optimizer (Loshchilov & Hutter, 2019), a cosine learning-rate scheduler with warmup, a peak learning rate of $3 \times 10^{-4}$, a global batch size of 1M tokens, and a context length of 1024. Since input embeddings in the language model contain no positional information (Dufter et al., 2022), we analyze the representations from the input embeddings up to the $Q^{(2)}K^{(2)\top}$ in order to examine how position-dependent attention score patterns emerge. Figure 3 presents the Gram matrix heatmap of inner products of the attention intermediates. The attention patterns from later layers are shown in Appendix Figure 8. For those figures, we used 1,000 snippets sampled from a held-out set drawn from Fineweb-Edu, each consisting of the first 50 tokens of a document.

First, because most embeddings are nearly orthogonal to one another, their inner products after LayerNorm are close to zero except along the main diagonal (a). This corresponds to the case $\alpha = 0$ in our earlier formulation. According to the simulation results in the previous section (Figure 2), under this condition the position-dependent patterns in the second-layer attention scores should appear only faintly. In contrast, the trained model exhibits a much stronger, $i$-independent pattern in $Q^{(2)}K^{(2)\top}$ (i), which more closely resembles the $\alpha \neq 0$ case. This is due to the learned parameters affect the position-dependent pattern induced by the causal mask.

Moving from (a) to (b), we observe that the ratio of off-diagonal to diagonal values increases substantially. This effect is attributed to the influence of $W_Q$ and $W_K$, as have a similar effect to

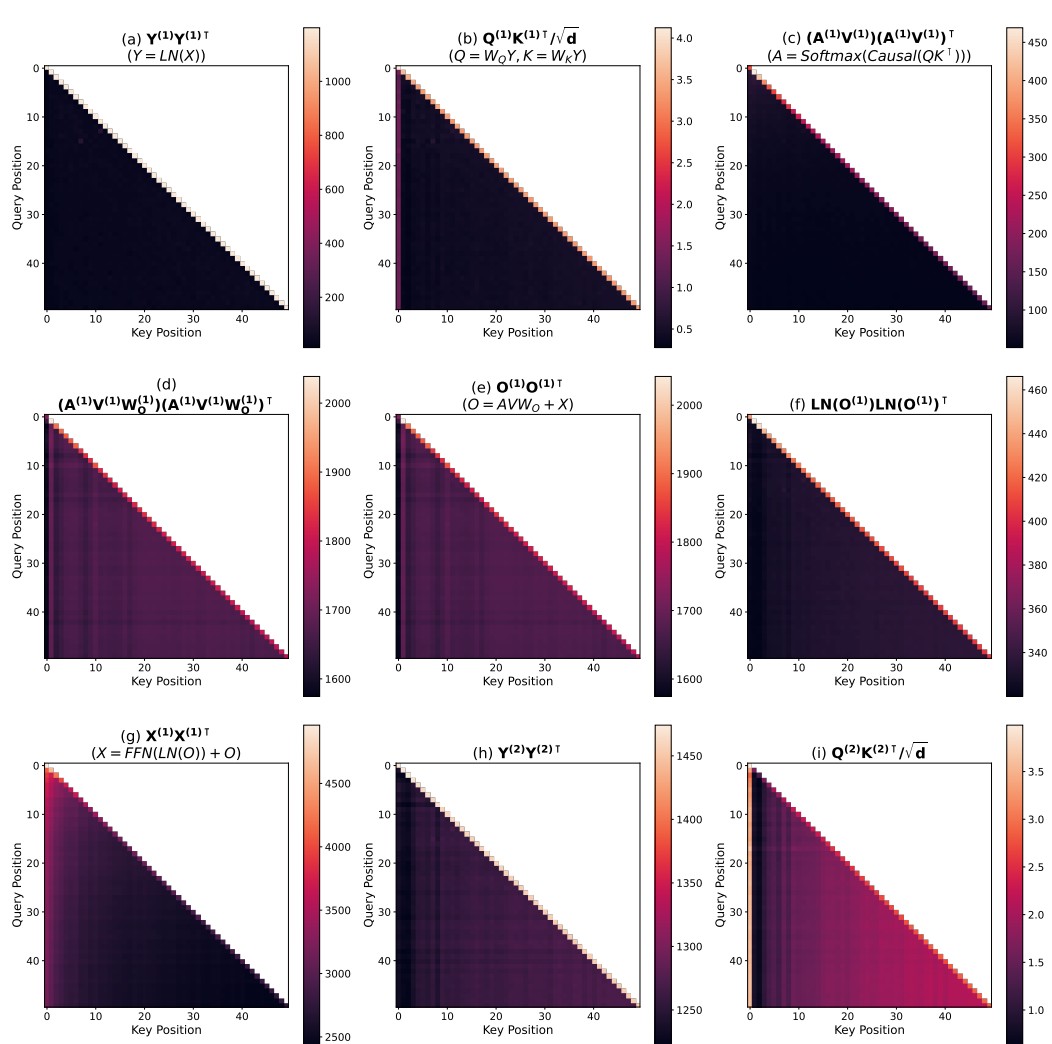

Figure 3: Inner-product Gram matrix heatmap of a trained Transformer decoder without positional encoding. We sample 1,000 sequences of length 50 from the held-out set, compute attention intermediates, and then calculate averaged inner products across heads and samples.

increasing $\alpha$. Although a vivid line appear at the 0th column, it can be interpreted as an attention sink phenomenon (Xiao et al., 2024), the analysis of which is beyond the scope of this paper.

After applying the causal mask, the softmax, and multiplying by $V$, the resulting attention map in (c) shows that the main diagonal values decrease with increasing $i, j$, while the off-diagonal values remain nearly uniform across positions. The diagonal corresponds to $h(i)^2$ with the residual term removed, while the off-diagonal corresponds to $g(i)$ with the residual term removed. We denote these by $h'(i)$ and $g'(i)$, respectively. Both can be computed in a manner similar to $h(i)$ and $g(i)$, and they exhibit analogous properties (Appendix A.2). In particular, $h'(i)$ decreases strictly with $i$, accounting for the decline along the diagonal, whereas $g'(i)$ is nearly independent of $j$ and only weakly dependent on $i$, explaining the uniformity of the off-diagonal elements.

In case of (d), the ratio between the diagonal and off-diagonal elements is significantly increased compared to (c). This can also be interpreted as the effect of $W_O$, analogous to the relationship between (a) and (b).

After adding the residual (e), the trend is almost identical to (d). This behavior can be explained in terms of $h(i)$ and $g(i)$.

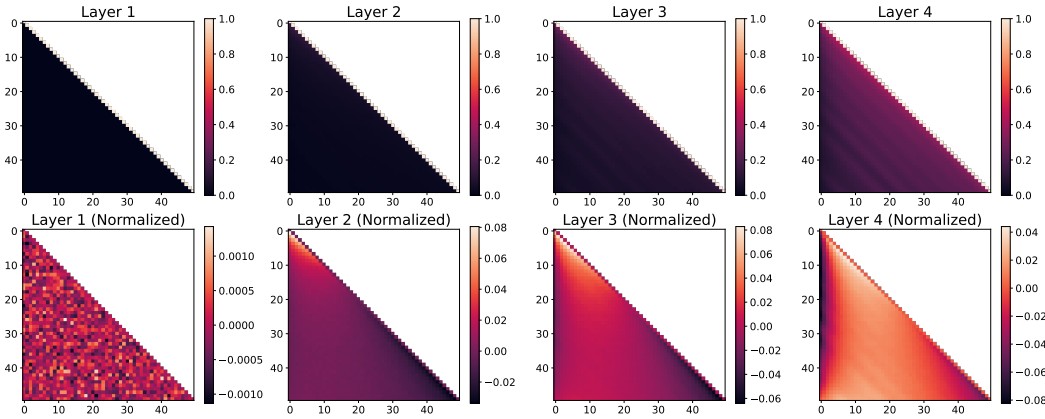

Figure 4: Simulation of a Transformer without parameters using RoPE. We visualize the attention scores across layers, where the y-axis denotes query indices and the x-axis denotes key indices. The top row shows the raw attention scores, while the bottom row shows scores normalized by subtracting the mean of each diagonal.

Once LayerNorm is applied (f), the pattern closely matches the theoretical and simulation results, where attention increases strictly with $j$ for fixed $i$.

Passing through the feedforward network and applying the residual (g) appears to laterally invert the pattern, whereas a subsequent LayerNorm restores it (h). Since we excluded the feedforward network from our theoretical analysis, we cannot account for why it inverted the relationship, nor for how LayerNorm reversed it. However, note that the position-dependent pattern has already emerged at (f), and therefore does not conflict with our theoretical analysis.

Finally, in the second-layer self-attention (i), we clearly observe the expected pattern of strict increase with $i$, in agreement with our theoretical analysis.

Although the behavior of (g) and (h) remains unexplained, our analysis shows how positional information emerge from a trained Transformer without explicit positional encoding. Also, we confirmed that the positional patterns induced by the causal mask are strongly parameter-dependent.

## 5 INTERACTION BETWEEN CAUSAL MASK AND RoPE

In the previous section, we showed that the causal mask can induce position-dependent attention patterns. Building on this, we now analyze how RoPE, widely used in modern LLMs, interacts with the causal mask. We first use simulations to examine the attention patterns that emerge from this interaction, and then check whether the same patterns appear in modern LLMs.

### 5.1 TRANSFORMER SIMULATION WITHOUT PARAMETERS

We extended our earlier simulation in Section 4.1 by applying RoPE, with $\theta_{\mathrm{RoPE}} = 10000$ and $\alpha = 0$. The results with non-zero $\alpha$ are provided in Appendix Figure 9. As shown in the first row of Figure 4, the first layer behaves as described by Barbero et al. (2025): when the input is independent Gaussian noise, RoPE alone does not affect the inner products. However, from the second layer onward, a vivid relative pattern emerges due to the mixing of the input vectors caused by the first layer's self-attention operation. By the third layer, this relative pattern becomes even clearer. Notably, similar to our earlier experiments without explicit positional encoding, the left portion of the attention maps appears darker than other regions. To highlight this effect, we averaged attention score along each diagonal and subtracted them and displayed on the second row, which revealed the pattern even more distinctly. Importantly, because RoPE is a relative positional encoding, such patterns do not appear without a causal mask (as in Transformer encoders; see Appendix Figure 10). This indicates that the causal mask biases the relative attention pattern from RoPE toward a non-relative one.

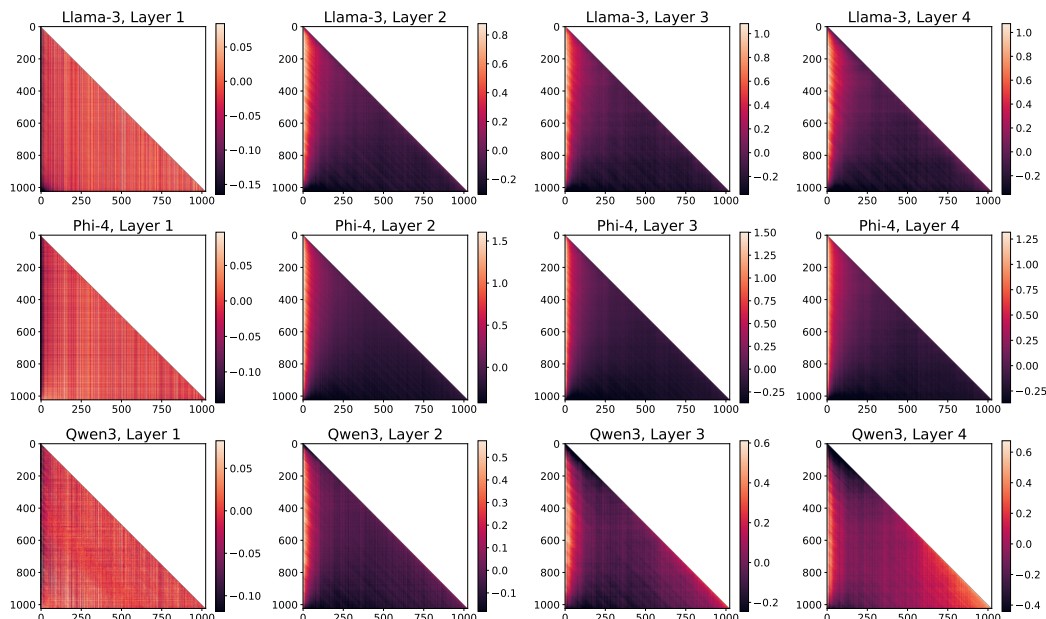

Figure 5: Diagonal-normalized attention heatmap of LLMs (first 4 layers). Attention scores were computed for 1,000 sequences of length 1,024 and averaged across sequences and heads. The Attention Sink effect flattens the overall pattern, so the color scale is adjusted using the 1% and 99% quantiles. The y-axis denotes query indices, and the x-axis denotes key indices.

## 5.2 ANALYSIS OF LLMS

We analyze whether the same phenomenon is observed in modern LLMs trained with RoPE, including Llama-3.1 8B Grattafiori et al. (2024), Phi-4 (Abdin et al., 2024), and Qwen3-8B (Yang et al., 2025a). Using the same setup as done at section 4.2, we performed inference on 1,000 samples from the same Fineweb-Edu (Penedo et al., 2024) held-out set, except increased sample length of 1024. Figure 5 shows the averaged attention scores for the first four layers of each model. Additional results, including the normalized and original scores for all layers, are provided in Appendix E. Consistent with our simulation, we can observe the non-relative pattern among the models except for the first layer. Also, considering the typical attention score of these models are in $[-10^1, 10^1]$ scale (Appendix E) and the non-relative patterns in $[-1, 1]$ scale, this effect could not be dismissed as negligible. This confirms that the attention patterns in practice are influenced by both RoPE and the causal mask, meaning that the Transformer Decoder trained with RoPE both relies on the positional information from RoPE and the causal mask. Since this pattern non-uniformly emphasizes the first few keys, it may potentially negatively affect length generalization.

## 6 CONCLUSION

Positional information is crucial for both the performance and length generalization of Transformer decoder–based models. While positional encodings are typically regarded as the primary source of positional information, the causal mask also conveys such information. In this work, we show how the causal mask encodes positional information and that its behavior is closely tied to common positional encodings. We further show that the causal mask influences the positional information derived from RoPE. Our study, however, has two key limitations. First, although Transformer decoder layers include feed-forward networks and learnable parameters, their effects have been only limitedly analyzed. Second, it remains unclear whether the interaction between RoPE and the causal mask directly affects performance and length generalization. Overall, our findings highlight the causal mask as a critical source of positional information, alongside explicit positional encodings such as RoPE. Exploring methods to jointly leverage the causal mask and explicit positional encodings may yield additional gains in LLM performance and length generalization.

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

# A ADDITIONAL DERIVATIONS

## A.1 STRICTLY DECREASING OF $h(j)$ ON $j$

From Equation 3

$$h(j)^2 = \frac{(2e + (j-1)e^\alpha)^2 + 2(2e + (j-1)e^\alpha)e^\alpha\alpha(j-1) + e^{2\alpha}(j-1)(1 + (j-2)\alpha)}{(e + (j-1)e^\alpha)^2}$$

$$
h(j+1)^2 - h(j)^2 = \frac{\begin{aligned}&\big((2e + je^\alpha)^2 + 2(2e + je^\alpha)e^\alpha\alpha j + e^{2\alpha}j(1 + (j-1)\alpha)\big)(e + (j-1)e^\alpha)^2\\ &- \big((2e + (j-1)e^\alpha)^2 + 2(2e + (j-1)e^\alpha)e^\alpha\alpha(j-1) + e^{2\alpha}(j-1)(1 + (j-2)\alpha)\big)\\ &\times (e + je^\alpha)^2\end{aligned}}{(e + je^\alpha)^2(e + (j-1)e^\alpha)^2}
$$

$$
= \frac{\begin{aligned}&\big((\alpha-1)e^{4\alpha} + 2(\alpha-1)e^{3\alpha+1}\big)j^2 + \big((2\alpha-6)e^{\alpha+3} + 2(\alpha-1)e^{3\alpha+1} + (\alpha-1)e^{4\alpha}\big)j\\ &+ (2\alpha-4)e^{\alpha+3} + (-2\alpha+4)e^{2\alpha+2}\end{aligned}}{(e + je^\alpha)^2(e + (j-1)e^\alpha)^2}.
$$

Since $0 \le \alpha < 1$, each coefficient (of $j^2$, $j$, and the constant term) is negative. The denominator is trivially positive, hence $h(j+1)^2 - h(j)^2 < 0$, so $h(j+1) - h(j) < 0$ ($\because h(j) > 0$). Thus, $h(j)$ is strictly decreasing on $j$.

## A.2 $h'(i)$ AND $g'(i)$

Without residual network, a single layer $f(X)$ acts as:

$$f(X) = \text{Softmax}(\text{Causal}(YY^\top))Y$$

Accordingly,

$$x_i^{(1)} = \frac{\sum_{k=1}^{i} f(i,k)x_k^{(0)}}{e + (i-1)e^\alpha} = \frac{ex_i^{(0)} + \sum_{k=1}^{i-1} e^\alpha x_k^{(0)}}{e + (i-1)e^\alpha}$$

In a same manner with above calculation, we can finally get

$$
\begin{aligned}
g'(i) &= \langle x_i^{(1)}, x_j^{(1)} \rangle\\
&= \frac{e\alpha + e^\alpha(1 + \alpha(i-2))}{e + (i-1)e^\alpha}\\
h'(i)^2 &= \langle x_i^{(1)}, x_i^{(1)} \rangle\\
&= \frac{e^2 + 2e^{1+\alpha}\alpha(i-1) + e^{2\alpha}(i-1)(1 + \alpha(i-2))}{(e + (i-1)e^\alpha)^2}
\end{aligned}
$$

We also can confirm $h'(i)$ is strictly decreasing on $i$, by showing

$$h'(i+1)^2 - h'(i) = \frac{(\alpha-1)e^\alpha(2e^3 - 2e^{2+\alpha}(i-1) + e^{3\alpha}i(i-1))}{(e + ie^\alpha)^2(e + (i-1)e^\alpha)^2} < 0.$$

## B EXTENDED CAUSAL MASK POSITIONAL INFORMATION PATTERN ANALYSIS

First, Figure 6 displays extension of Figure 2 with additional $\alpha$ values. With higher $\alpha$ value, the pattern faster saturate rapidly to a constant on $j$.

Next, we conducted the same experiment using LayerNorm with $\sqrt{d}$ scaling, instead of $\ell_2$ normalization without scaling. The results are shown in Figure 7. Since LayerNorm normalizes vectors to have a norm of $\sqrt{d}$, the inner product after $\sqrt{d}$ scaling becomes $\sqrt{d}$ times larger than with $\ell_2$ normalization. This makes the softmax distribution sharper than in the $\ell_2$ case, thereby reducing the influence of positional information introduced by the causal mask. However, simply changing the scaling factor from $\sqrt{d}$ to $d$ recovers behavior similar to $\ell_2$ normalization, validating our findings.

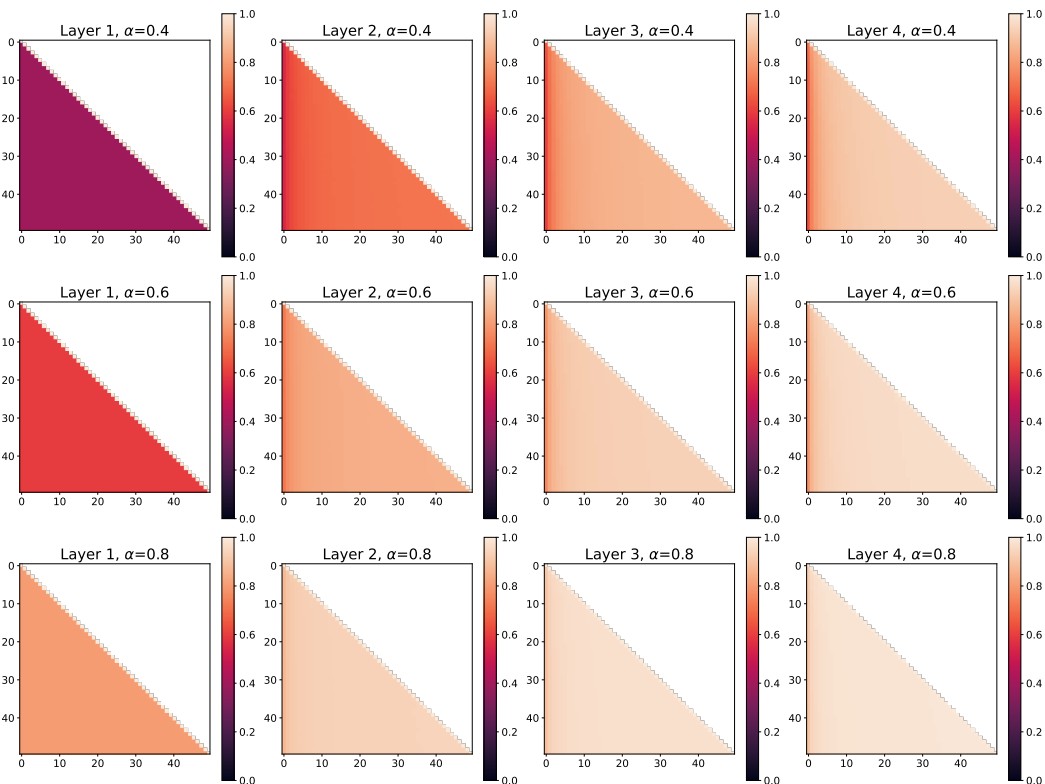

Figure 6: Extended result of Figure 2.

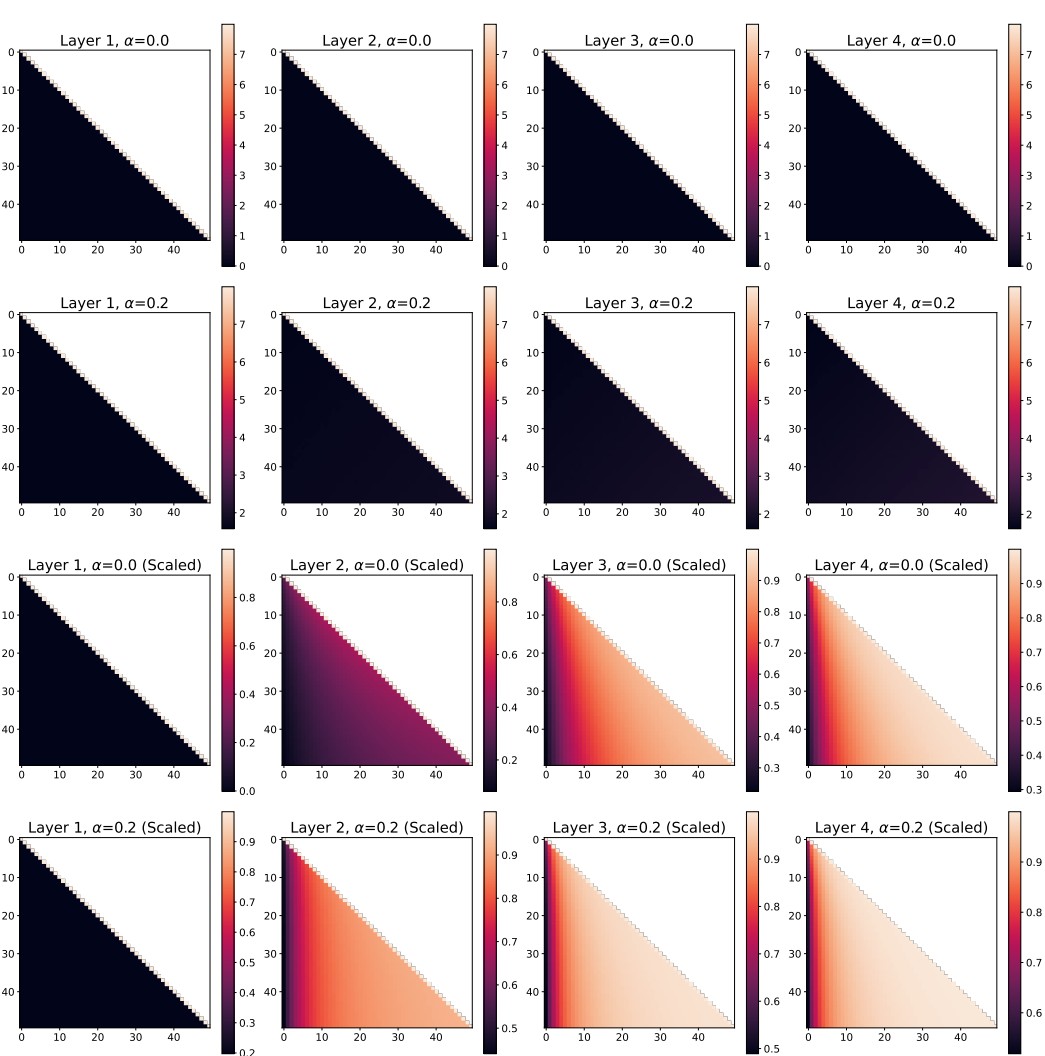

Figure 7: Simulation of a Transformer without parameter and explicit positional encoding results, with LayerNorm. We replaced $\ell_2$ normalization to LayerNorm, and applied $\sqrt{d}$ scaling (first and second row) and $d$ scaling (third and fourth row).

## C ATTENTION PATTERN OF A TRAINED TRANSFORMER WITHOUT EXPLICIT POSITIONAL ENCODING

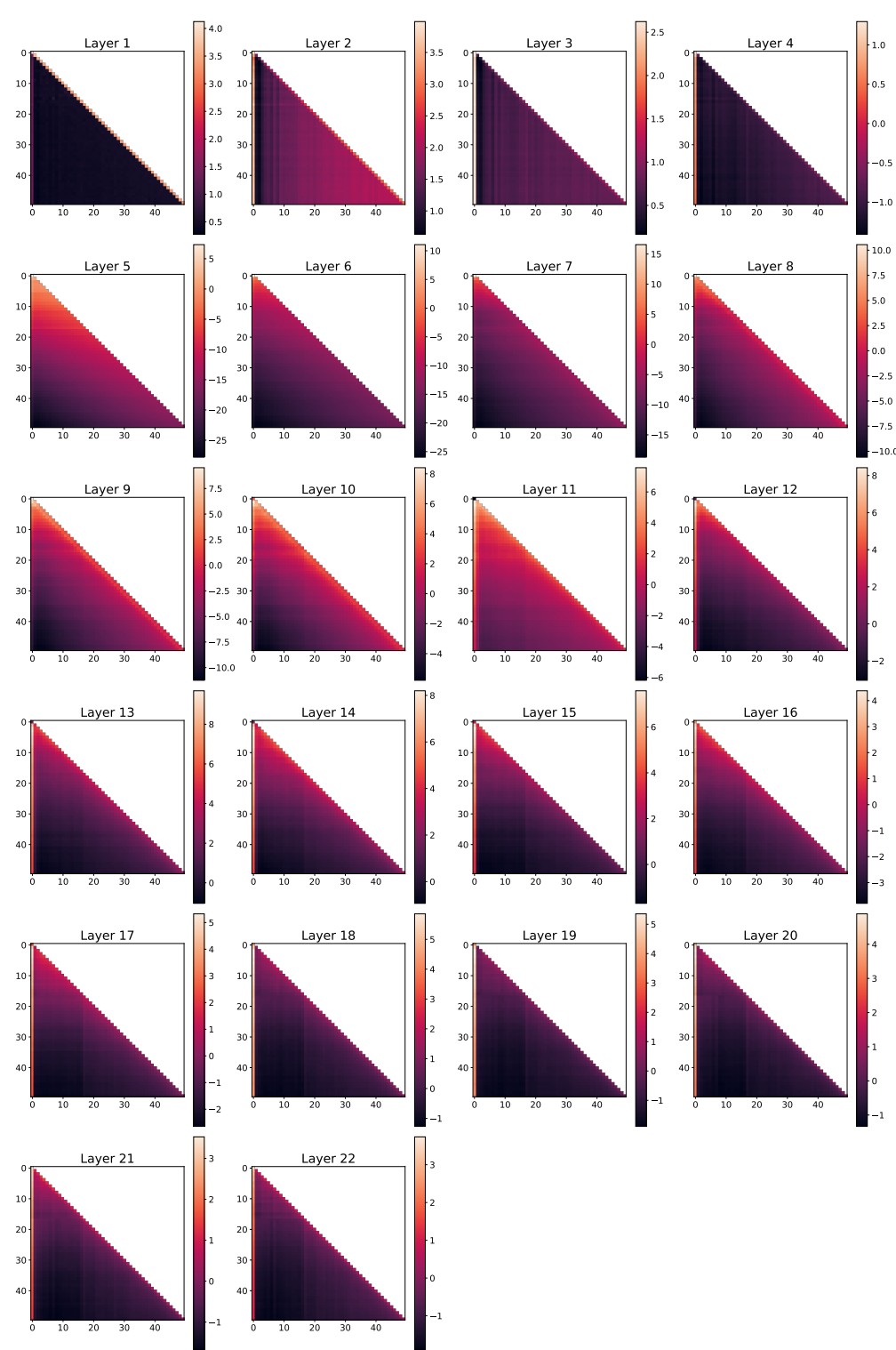

Figure 8: Attention pattern of a trained Transformer without explicit positional encoding. The plots are drawn with a same manner to Figure 3.

## D    RoPE Attention Pattern Analysis Results

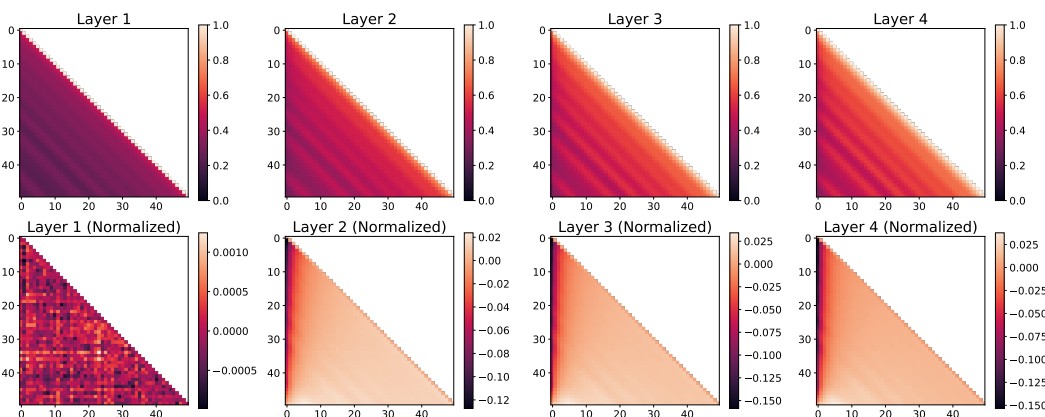

Figure 9: The extended result of Figure 4 with $\alpha = 0.5$

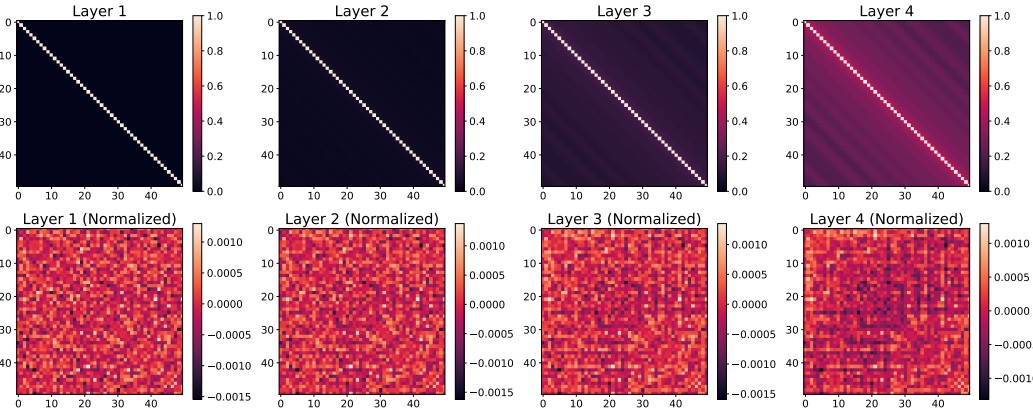

Figure 10: The extended result of Figure 4 without causal mask (i.e. Transformer Encoder)

## E    LLM RoPE Patterns

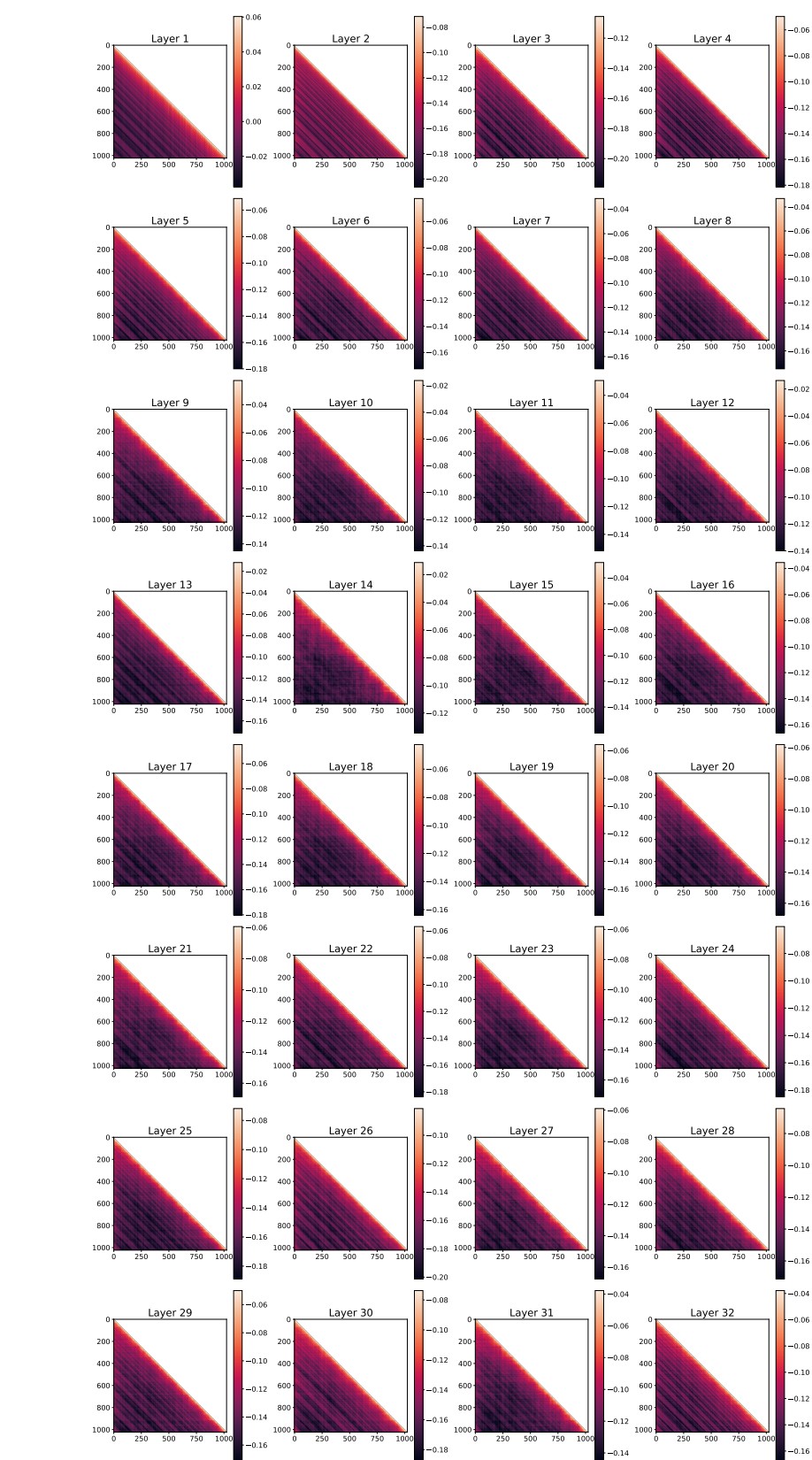

Figure 11: Llama-3-8B Attention Pattern

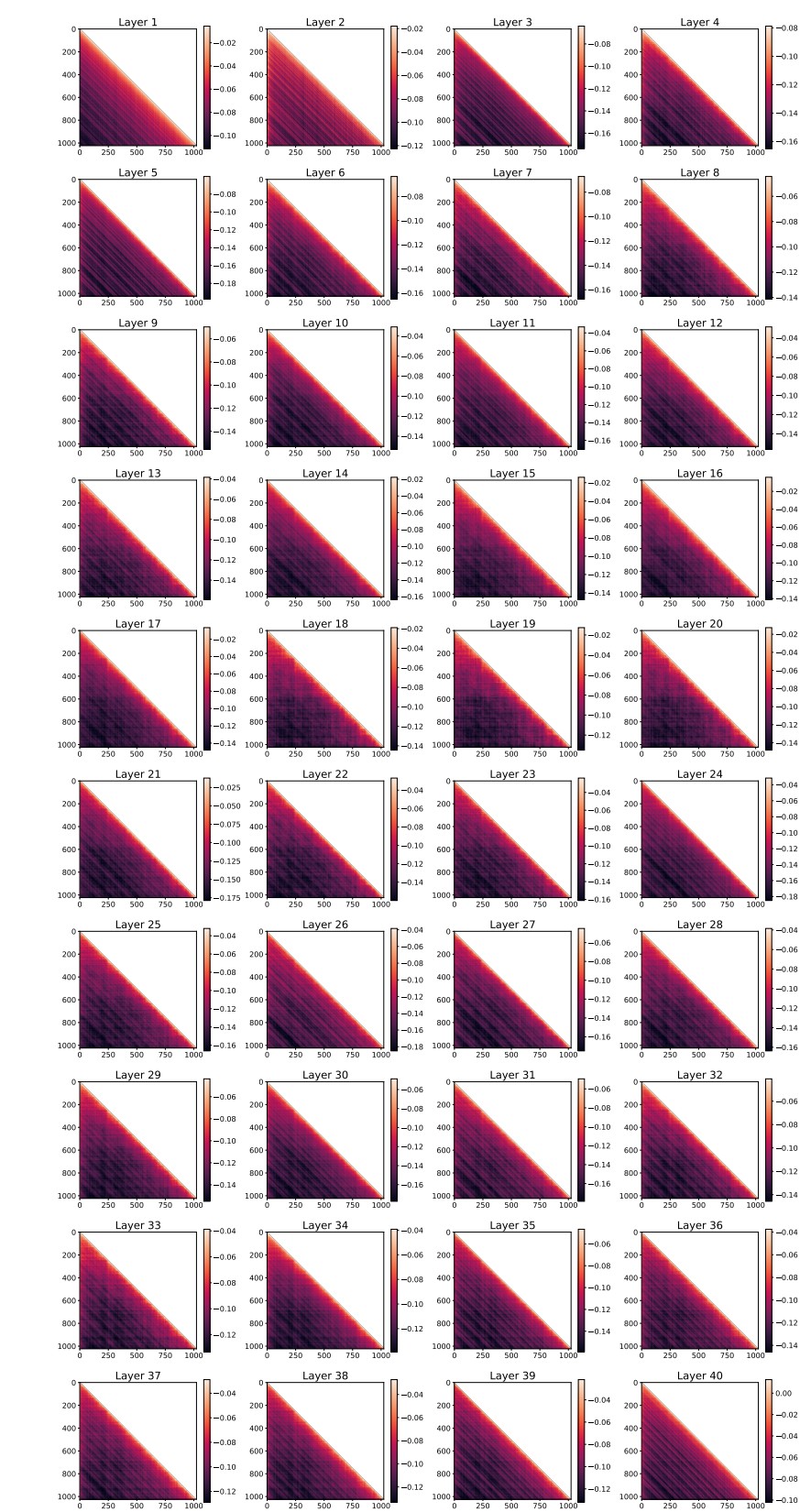

Figure 12: Phi-4 Per-Layer Attention Pattern

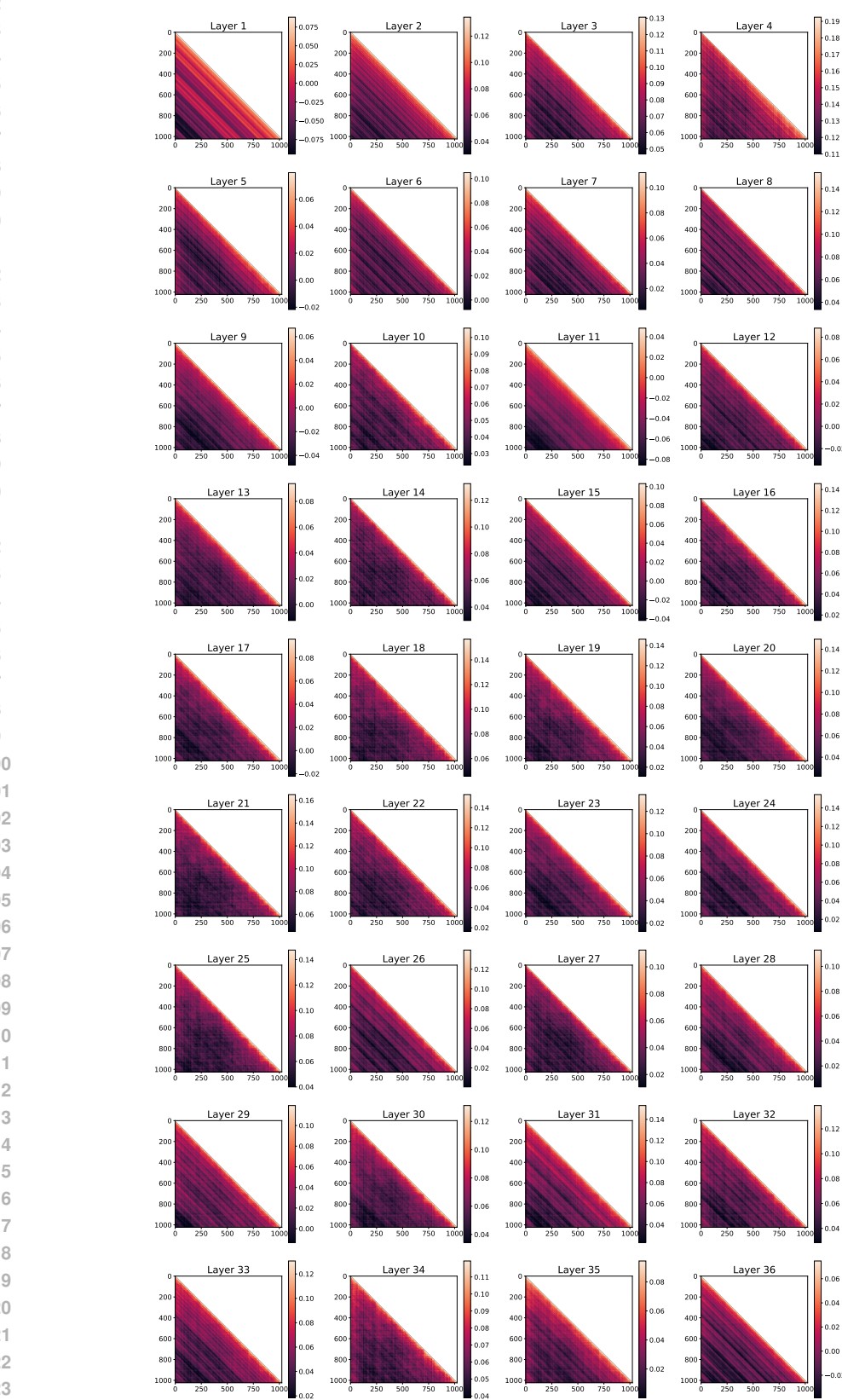

Figure 13: Qwen3-8B Per-Layer Attention Pattern

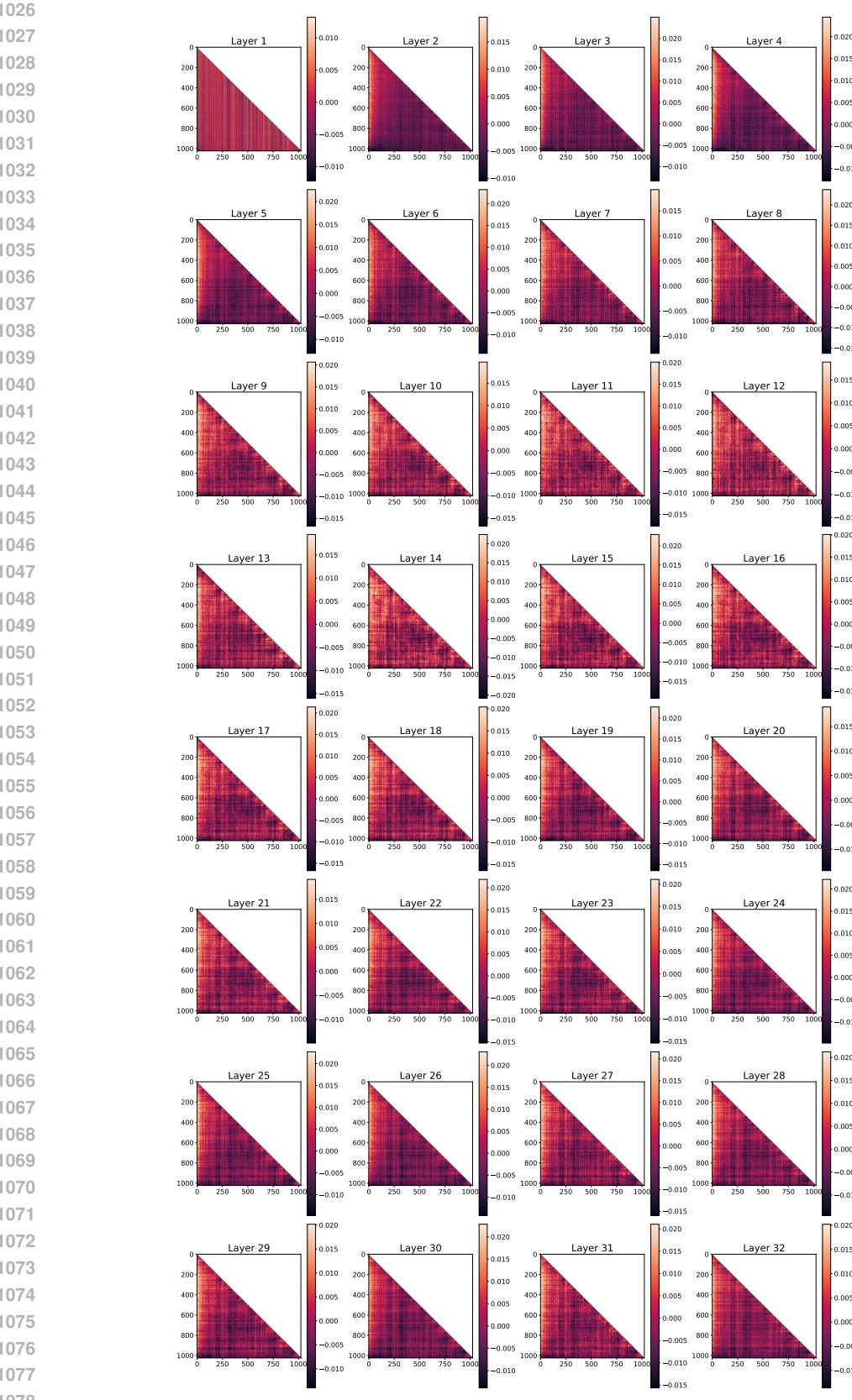

Figure 14: Llama-3-8B Attention Pattern (Normalized)

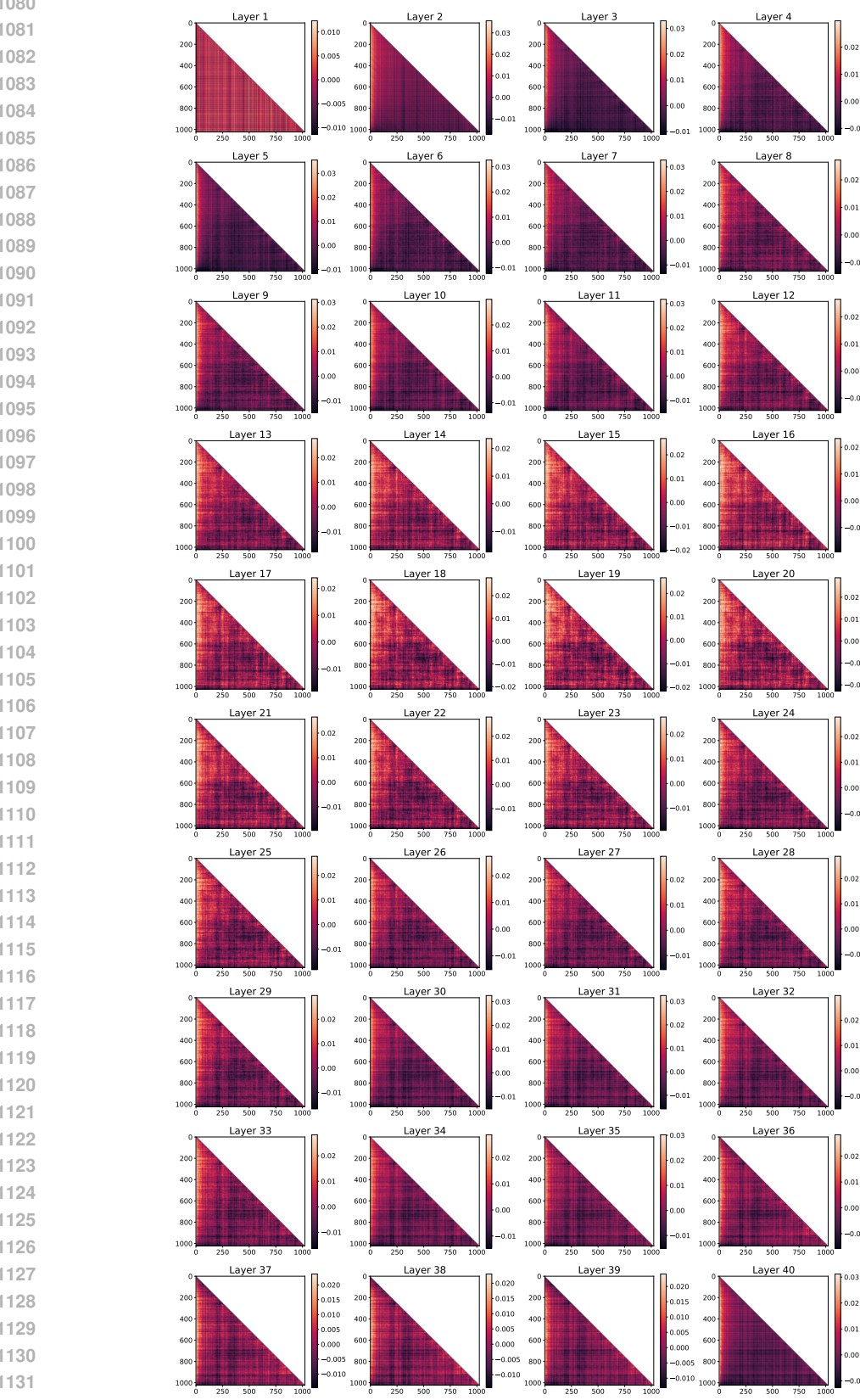

Figure 15: Phi-4 Attention Pattern (Normalized)

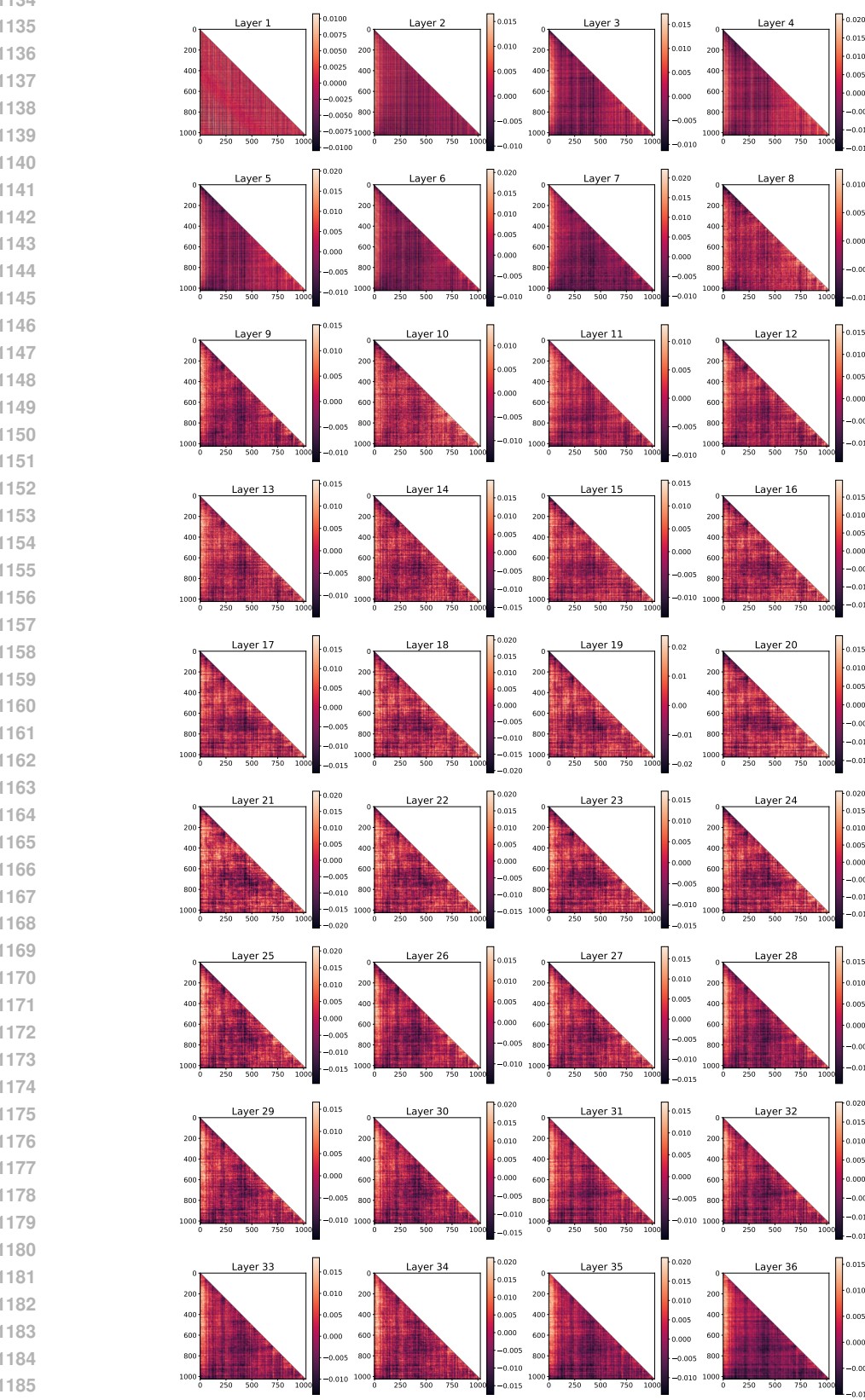

Figure 16: Qwen-8B Attention Pattern (Normalized)

