# OpenReview forum: "Behind RoPE: How Does Causal Mask Encode Positional Information?"
_ICLR.cc/2026/Conference — ICLR 2026 Conference Withdrawn Submission_

### Official Review · Reviewer_Pzuo · 2025-10-20

**Soundness:** 2
**Presentation:** 2
**Contribution:** 1
**Rating:** 2
**Confidence:** 4

**Summary:**

This paper analyzes how the causal mask in Transformer decoders encodes positional information, beyond the commonly studied explicit positional encodings like RoPE. As for the **theoretical analysis**, it:
- Proves that causal masks induce position-dependent attention patterns even without parameters, causal input dependencies, or feedforward networks
- Shows these patterns favor nearby query-key pairs (higher attention scores for closer positions), similar to explicit positional encodings
- Demonstrates the mechanism works through cumulative effects across layers, where position-dependent patterns emerge more clearly in deeper layers

In terms of **empirical analysis**, it:
- Simulates parameter-free Transformers confirming the theoretical predictions
- Trains a 1.5B parameter model without positional encoding on 20B tokens, showing learned parameters amplify the causal mask's positional patterns
- Analyzes attention patterns in modern LLMs (Llama-3.1-8B, Phi-4, Qwen3-8B)

For **interaction with RoPE:**

- Discovers that causal masks distort RoPE's relative attention patterns into non-relative ones
- Shows this distortion creates position-dependent biases that emphasize earlier tokens
- Observes this phenomenon consistently across modern LLMs at non-negligible scales

**Strengths:**

1. It provides a clear mathematical explanation for a proof that causal masks alone can encode positional information, even without any parameters or explicit positional encodings.
2. It reveals a mechanism affecting every Transformer decoder, showing that causal masks create position-dependent attention patterns that favor nearby tokens and interact with RoPE to distort its relative patterns into non-relative ones.
3. It proposes empirical analysis in models like Llama and Qwen.

**Weaknesses:**

My major concern is the **novelty** of this paper. For a paper submitted to a top-tier conference as a contribution to the entire LLM community, novelty, i.e., studying a topic that has not been discussed or solved before, is of paramount importance. However, regarding the major contributions of this paper (which I summarized below), most of them have been proved theoretically or validated experimentally in previous works:

- **Causal mask itself encodes positional information**: This has been extensively studied in "Transformer Language Models without Positional Encodings Still Learn Positional Information" (https://arxiv.org/abs/2203.16634), "Why Are Positional Encodings Nonessential for Deep Autoregressive Transformers? Revisiting a Petroglyph" (https://aclanthology.org/2025.findings-acl.30.pdf), "StableMask: Refining Causal Masking in Decoder-only Transformer" (https://arxiv.org/abs/2402.04779) Section 4.2, and "On the Emergence of Position Bias in Transformers" (https://arxiv.org/abs/2502.01951) Theorem 4.1, Section 5.2.

- **Interaction between RoPE and causal mask**: This phenomenon has been analyzed in "Your Transformer May Not be as Powerful as You Expect" (https://arxiv.org/abs/2205.13401), and more comprehensively in "On the Emergence of Position Bias in Transformers" (https://arxiv.org/abs/2502.01951) Theorems 4.5-4.7, Section 5.1.

- **Empirical validation**: Similar experimental approaches have been employed in "Transformer Language Models without Positional Encodings Still Learn Positional Information" and "Why Are Positional Encodings Nonessential for Deep Autoregressive Transformers? Revisiting a Petroglyph."

Some minor points:

- Attention heat map is not a good validation of the model's actual behavior. As for validating the role of positional encoding, I suggest some toy tasks e.g., repeat some numbers with absolute / relative positions.

- The experimental validation, while comprehensive in scope, largely confirms what previous works have already established. The analysis of modern LLMs (Llama, Phi, Qwen) demonstrates the universality of these effects but does not advance our understanding beyond existing literature.

**Questions:**

1. Please provide additional experiments mentioned in the weaknesses section.
2. Can authors explain what is new in this paper, considering a large number of related works have discussed this topic with similar analysis and conclusions?

---

### Official Review · Reviewer_NDF8 · 2025-10-28

**Soundness:** 1
**Presentation:** 2
**Contribution:** 1
**Rating:** 2
**Confidence:** 4

**Summary:**

The authors argue that the causal mask can induce position-dependent patterns in attention scores. The authors then provide supporting empirical and mathematical results.

**Strengths:**

The paper is well written and clear.

**Weaknesses:**

My main concern is that the main contribution has already been explored. For instance, [1] shows "causal
masking inherently biases attention toward earlier positions" and in section 4.3 they also investigate in detail how this combines with RoPE. Another example is [2] in which this Positional bias due to the causal mask has also been pointed out. I don't see these works being discussed even though they seem extremely relevant.

The abstract also claims "prove that the causal mask can induce position-dependent patterns in attention scores", but "prove" to me here seems like a strong word. It would be important to turn the calculation in page 4 as a precise mathematical statement to understand better what the precise claim is there.

[1] On the Emergence of Position Bias in Transformers. ICML 2025
[2] Transformers need glasses! Information over-squashing in language tasks. NeurIPS 2024

**Questions:**

Would the authors be able to explain how their work differs from the works mentioned in the weaknesses, especially [1]?

Please see other weaknesses.

Overall, while I found the paper to point in an interesting direction, I do not believe the mathematical results to be rigorous enough and importantly I do not see novel contributions given the referenced papers.

---

### Official Review · Reviewer_jexM · 2025-10-31

**Soundness:** 3
**Presentation:** 3
**Contribution:** 2
**Rating:** 4
**Confidence:** 2

**Summary:**

The paper shows that the causal mask influences the positional information. It analyzes how simplified transformer (transformers without learnable parameters and FFN) could extract positional information using causal mask only.
Authors show that  attention score is not constant across query-key indices for simplified model. On top of this also is shown that attention score in the second layer strictly increases on the key index j ≤ i over fixed query index i. derivations are detailed for simplified model
It also does empirical analysis of simplified model, simplified model with RoPE and trained models with NoPE.

**Strengths:**

- Detailed derivation of attention scores for simplified transformer model (transformers without learnable parameters and FFN)
- Extensive empirical analysis

**Weaknesses:**

- limited novelty. The fact that NoPE + Casual masking could represent positional information  is well known at least since the work of Kazemnejad, et. al
- transformer model without learnable parameters and FFN is significant simplification.  Even though 4.2 empirically shows similarity of the patterns learned between model trained without positional encoding and simplified model, link is not very well established.

**Questions:**

line 165. Could you please clarify, what do you mean exactly by "upper triangular matrix"? standard definition implies zeros below the main diagonal. Here we have alpha instead.

nit: line 234: "Transformer without parameter"  . Should it be "parameters"?

---

### Official Review · Reviewer_yrEE · 2025-11-01

**Soundness:** 2
**Presentation:** 3
**Contribution:** 2
**Rating:** 2
**Confidence:** 2

**Summary:**

The paper is trying to elucidate how causal masking leads to positional encoding, or at least to attention weights that decrease with distance. The claim is that this happens even in the absence of specifically learnt weights, and is a natural outcome of the behaviour of the softmax normalization. This is first described mechanically by showing the mathematical machinery behind the behaviour. Then a bunch of empirical evidence is provided by looking at models trained without and with RoPE encoding. The paper concludes with the remark that causal masking should not be ignored in any study involving positional encoding or length generalization.

**Strengths:**

Transformers are inherently architectures that work on sets, though are mainly used in sequential modelling. To bridge this gap between sets and sequences, positional encoding plays a crucial role, providing the order of the elements. Therefore understanding how positional information evolves in the architecture is crucial. This is particularly so in the light of recent work that bring questions about some conventional methods of providing this information, as for example RoPE and empirical evidence that models work on sequential data surprisingly well even when the explicit positional encoding information is not provided.

All in all, I think the topic of the paper is very timely, and indeed we have a poor understanding of how the causal mask helps in relating this information. The work points towards an inherent bias of the attention mechanisms (compared to previous works that argue that there exists weights such that jointly with causal mask you get positional information). This seems a stronger claim and therefore interesting for the community.

**Weaknesses:**

While the paper reads well, I feel like the authors are not very explicit about the meaning of their derivations and empirical evidence. Specifically I feel like the authors should be a bit more explicit which quantities are basically an expectation (or an approximation of an expectation). E.g. for clarity, it is E[ <x_i, x_j> ] = alpha.  Therefore in 3.2, e.g. in eq 3, we are looking at expected behavior, when we simplify that <x_i, x_j> = alpha right? But the softmax is applied over specific instances, not over expected values. I wonder if the issue comes from eq.2  which again is not clear to me if it suppose to represent the expected value or the instance value for some given x_i, x_j.  Maybe it would be useful for the authors to say more exactly how we should interpret the mathematical derivation.

I can imagine that the dimension over which the reduction is being made is d, i.e. the dimension encoding the length of the embedding, and the assumption is that every entry on the embedding is an i.i.d. sample. But this is a strange assumption to make, which is obviously not true in practice. And this is definitely not what the notation on Figure 1 is suggesting.

I think the empirical evidence also shows average attention maps over many samples, which continue to build this intuition that what we are talking about is that causal masking in expectation leads to a form of positional encoding.

But my issue is that I do not understand why the expected behavior is meaningful here. Positional encoding is relevant to the architecture to the extent it can be used in the processing of a given instance. However neither the empirical evidence nor the math suggest that for a given instance of a random input sequence you can still decode any kind of positional information due to the causal masking alone (i.e. removing any other kind of positional encoding). For this to happen for any given sequence (in a per sequence scenario), I guess you still need to learn specific weight matrices. I do wonder if the real claim is that this expected behavior acts as an inductive bias, that allows learning to discover these kind of weights that allows to resolve positional encoding on a per instance basis. But this claim is not really made in the paper either.

I admit that I might misunderstood some fundamental aspect of the paper, and I hope the authors can explain my misunderstanding and improve the write-up such that future readers don't struggle with the same issues. But if I did not misunderstood it, then my worry is that the paper does not really provide a functioning mechanism through which causal masking encode position -- or at least not one that can be exploited for a given instance by the transformer.

**Questions:**

1. We are making an assumption on the expected value of <x_i, x_j>,  i.e. E[x_i x_j] = alpha,  but unless I'm misunderstanding the text, eq 2. drops the expectation and assumes  that  x_i x_j = alpha, otherwise I'm confused by the derivation. Should I assume that the entire derivation looks at these quantities in expectation?

2. Can you comment of whether the paper discusses about the expected behavior of the attention weights, or does it provide insight in how the causal masking caries positional information in any given instance that can readily be exploited by the transformer in how it attends?

3. Would learning weights that allow embeddings to contain semantical information somehow interfere with this mechanism (e.g. this has been brought up against RoPE, that it interferes with semantical information). Is this an issue that we should consider here as well ?

4. More of a minor question. But the authors describe how the information encoded by causal masking doesn't behave as that of relative positional encoding (e.g. RoPE) nor is it behaving like absolute positional encoding (line 298-300). Can  you expand what are the implications of this? How should we intuitively think of these differences? What kind of encoding is this then. ?

**Details Of Ethics Concerns:**

No reason for ethical reviewing.

---

### Note · Authors · 2025-11-14

**Comment:**

Thank you very much for the valuable and constructive reviews. After carefully considering the feedback on novelty from reviewers NDF8 and Pzuo, as well as the mathematical insights from yrEE, we have decided to withdraw this paper.

**Withdrawal Confirmation:**

I have read and agree with the venue's withdrawal policy on behalf of myself and my co-authors.